# Magnetosomes and Magnetosome Mimics: Preparation, Cancer Cell Uptake and Functionalization for Future Cancer Therapies

**DOI:** 10.3390/pharmaceutics13030367

**Published:** 2021-03-10

**Authors:** Zainab Taher, Christopher Legge, Natalie Winder, Pawel Lysyganicz, Andrea Rawlings, Helen Bryant, Munitta Muthana, Sarah Staniland

**Affiliations:** 1Department of Chemistry, University of Sheffield, Brook Hill, Sheffield S3 7HF, UK; zamtaher1@sheffield.ac.uk (Z.T.); cjlegge1@sheffield.ac.uk (C.L.); njwinder1@sheffield.ac.uk (N.W.); pkl29@cam.ac.uk (P.L.); andrea.e.rawlings@gmail.com (A.R.); 2Department of Oncology and Metabolism, University of Sheffield, Beech Hill Road, Sheffield S10 2RX, UK; h.bryant@sheffield.ac.uk (H.B.); m.muthana@sheffield.ac.uk (M.M.); 3Department of Biomedical Sciences, University of Sheffield, Western Bank, Sheffield S10 2TN, UK

**Keywords:** magnetite, magnetosome, silica coating, MDA-MB-231, cell uptake, functionalization, Biotinylation

## Abstract

Magnetic magnetite nanoparticles (MNP) are heralded as model vehicles for nanomedicine, particularly cancer therapeutics. However, there are many methods of synthesizing different sized and coated MNP, which may affect their performance as nanomedicines. Magnetosomes are naturally occurring, lipid-coated MNP that exhibit exceptional hyperthermic heating, but their properties, cancer cell uptake and toxicity have yet to be compared to other MNP. Magnetosomes can be mimicked by coating MNP in either amphiphilic oleic acid or silica. In this study, magnetosomes are directly compared to control MNP, biomimetic oleic acid and silica coated MNP of varying sizes. MNP are characterized and compared with respect to size, magnetism, and surface properties. Small (8 ± 1.6 nm) and larger (32 ± 9.9 nm) MNP are produced by two different methods and coated with either silica or oleic acid, increasing the size and the size dispersity of the MNP. The coated larger MNP are comparable in size (49 ± 12.5 nm and 61 ± 18.2 nm) to magnetosomes (46 ± 11.8 nm) making good magnetosome mimics. All MNP are assessed and compared for cancer cell uptake in MDA-MB-231 cells and importantly, all are readily taken up with minimal toxic effect. Silica coated MNP show the most uptake with greater than 60% cell uptake at the highest concentration, and magnetosomes showing the least with less than 40% at the highest concentration, while size does not have a significant effect on uptake. Finally, surface functionalization is demonstrated for magnetosomes and silica coated MNP using biotinylation and EDC-NHS, respectively, to conjugate fluorescent probes. The modified particles are visualized in MDA-MB-231 cells and demonstrate how both naturally biosynthesized magnetosomes and biomimetic silica coated MNP can be functionalized and readily up taken by cancer cells for realization as nanomedical vehicles.

## 1. Introduction

Nanoparticle (NP)-based medicines involve the use of nanoscale materials, and they can be used in the diagnosis and treatment of diseases [1,2]. The majority of NPs have focused on the use of particles as a delivery vector for drugs, as well as mRNA sequences for vaccine therapy [3]. Multiple materials have been utilized for this delivery method; liposomes involve the use of a lipid membrane to encapsulate the therapeutic for delivery, and more recently liposomes have been altered to be able to increase circulation time by creating stealth vesicles [4,5]. Polyethylene glycol can increase the circulation time of the particle, and this is beneficial for several reasons as increased retention can increase the chance of uptake of the particle into tumor cells [4,5,6]. Although liposomes have many advantages, there are distinct disadvantages, as liposomes require special storage conditions as well as being prone to oxidative damage [7,8].

Nanocarriers are also capable of delivery of drugs to areas of the body that are protected and can restrict the ability to deliver drugs to the target, such as the blood–brain barrier [9]. The blood–brain barrier is a semipermeable membrane that regulates the movement of ions and molecules, to cross into the central nervous system (CNS). This barrier can also restrict drugs from crossing into the CNS, making diseases that target the brain difficult to treat [10]. Nanocarriers can help to overcome the difficulties of delivering therapies to the brain, and niosomes are a class of NP that are effective in crossing the blood-brain barrier. Niosomes can be single and multilamellar composed of lipids and nonionic surfactants that can encapsulate both hydrophilic and hydrophobic drugs for delivery [9]. These NPs are generally hollow spherical particles that can be used as carriers for a therapeutic payload [11]. A further class of NP are iron oxide nanoparticles (IONP) and these are naturally occurring as well as synthetically produced. Not only do they offer the ability to be a nanocarrier, they are also steerable with an external magnetic stimuli as well as being able to directly treat tumor targets via magnetic hyperthermia [2,12].

The use of magnetic iron oxide NPs as a targeted treatment for cancer was proposed in the late 1970s and since then research in this area has significantly increased. MNP are a key component in the field of nanomedicine. MNP make the ideal drug delivery vehicle: The nanoscale size offers a large surface area to volume ratio to display therapeutics such as drugs and enables them to pass unhindered around the circulatory system [13]. MNP have the ability to be directed to a site of interest using a Magnetic Resonance Imager [12]. Further to drug delivery, magnetism can also be utilized for magnetic hyperthermic therapies [14].

Hyperthermia treatment involves the destruction of tumors by heating the tumor cells to temperatures in excess of 41 °C, which leads to permanent cellular damage [15]. The exposure of cancerous cells to hyperthermia conditions causes permanent protein degradation leading to loss of cellular function [16]; if temperatures exceed 60 °C, coagulative necrosis of the tumors can occur [17]. The attraction of thermoablative treatments over current surgical resection are many and include a lower risk of complications and shorter hospitalization, improving quality of life [18]. IONPs offer a unique method of delivering hyperthermia treatments to a tumor site due to their small size, as they can be delivered intravenously and directed to the site of interest using magnetic resonance imaging [12,19].

Magnetosomes are naturally occurring MNP that are composed of a magnetite core, within a phospholipid bilayer, biosynthesized within magnetotactic bacteria (MTB) [20,21,22]. The formation of the magnetic core is a process controlled by biomineralisation proteins within the magnetosome membrane, and these proteins control the size and shape of the MNP formed, resulting in highly uniformed particles [20,22,23]. MTB produce chains of magnetosomes, which are single domain MNP with a diameter of 35–120 nm depending on the strain of the MTB [22], with *Magnetospirillum magneticum* (AMB-1) MTB producing cubo-octohedral MNP of 45 nm diameter. Such magnetosomes have impressively high specific absorbance ratio (SAR) values of 960 W/g, significantly higher than SAR values observed from synthetically synthesized MNP [24,25]. Magnetosome chains extracted from AMB-1 have been shown to be very effective in hyperthermia experiments targeting breast cancer cell lines, showing greater efficiency in hyperthermia treatments when compared to other MNP assumedly due to a combination of the high SAR values of magnetosomes and their greater uptake potential [23,24,26].

Although magnetosomes show exceptional potential for nanomedicine over other types of MNP, they do suffer some drawbacks, namely biosynthesis within MTB is costly and slow with small quantities produced after a lengthy purification, making scale-up and large-scale use unfeasible [27]. The use of synthetic MNP can overcome these issues by production with quick, low-cost methods, but there is the trade-off with less than optimal physical properties and potentially lower uptake when compared to magnetosomes [23,28]. A key aspect to be retained from the natural process is the environmentally friendly processing. The use of magnetosome mimics have shown promise in these areas, displaying high SAR values (122) and have been been effective in hyperthermia treatment of cancer cells [2,29]. However, to date, there is no study that compares the uptake of magnetosomes, chemically produced MNP, and biomimetically coated MNP into cancer cells. The advantages and disadvantages of each are shown in Table 1).

Furthermore, a biocompatible coating present on both the magnetosomes and biomimetic MNP enables the attachment of targeting moieties or anti-cancer therapies [2,30]. The ability to conjugate proteins, drugs, and other compounds to the surface of magnetosomes and MNP greatly increase their value as biotherapeutics, allowing for their use as not just a therapeutic, but also a theranostic tool and steerable therapeutic delivery vehicle [12,31]. Conjugation of compounds to the surface of MNP is possible through a variety of methods, which have different advantages. Glutaraldehyde conjugation of antibodies has been shown to be an effective and fast conjugation method for antibody binding to the surface of functionalized MNP [2]. Biotin streptavidin binding is a fast and very specific method for conjugation and immobilization to functionalize NPs with proteins. EDC-NHS coupling allows for specific chemical binding of primary amine groups to carboxylic acid groups allowing for binding of MNP to the target compound without any non-specific crosslinking, both naturally occurring and chemically synthesized.

MDA-MB-231 cells are an estrogen, progesterone, and Her-2 receptor negative cell line that shows invasive tendencies in vitro and is a good model of late stage triple negative breast cancers (TNBCs). TNBCs are known to have a worse prognosis than other breast cancers due to the fewer treatment options, with chemotherapy being the only systemic treatment option [32,33,34]. The delivery of therapeutic agents directly to the tumor site would allow for lower dosage treatments improving quality of life and reducing the aggressiveness of the treatment.

In this study, we characterize and compare magnetosomes, synthetic MNP of different sizes, and coated biomimetic MNP for their uptake by MDA-MB-231 cells. The biocompatible magnetosome mimics are MNP coated with the amphiphilic oleic acid (OA@MNP) and with silica (Si@MNP) (Figure 1). We produce Si@MNP at two sizes to examine the effect of size as well as coating. Furthermore, we show that we can functionalize the MNP by stably conjugating compounds to the surface of both magnetosomes and the biomimic Si@MNP (Figure 1) and we compare the effectiveness of the different MNP types and conjugation methods.

## 2. Materials and Methods

### 2.1. Methods of Particle Preparation

#### 2.1.1. Bacteria Growth Condition

*Magnetospirillum* AMB-1 was grown in 400 mL of minimal media (DSMZ 380 medium), in 500 mL bottles with loose lids placed in a microaerobic cabinet at 30 °C with 99% nitrogen, 1% oxygen gas. 8 μL of 1 M ferric quinate added to give a final concentration of 20 μM iron. MTB growth rate was measured by optical density at 600 nm wavelength.

#### 2.1.2. Harvesting Cells

*Magnetospirillum* AMB-1 were concentrated by centrifugation (50 mL cell suspensions at 4700 rpm in a TX-750 rota for 45 min) and supernatant removed to leave the pellet. The pellets were combined and resuspended in 20 mL of Tris-HCl buffer (pH 7.4). The cells were lysed by tip sonication. A strong neodymium magnet was used to separate the magnetosomes from the cell debris. The magnetosomes were washed and magnetically extracted 5 times with 10 mM Tris-HCl and a final time sterile water. 

#### 2.1.3. Surface Functionalization of Magnetosomes: Biotinylation

Sulfo-N-hydroxylsuccinimide biotin (Sulfo-NHS-Biotin) and primary amino groups (lysine) found on the magnetosomes were chosen for biotinylation. Three mg of magnetosomes were washed 3 times with 100 µL (ice cold) HEPES buffer (20 mM pH 7.4) (Sigma Aldrich, St Louis, MO, USA), to remove amine groups contained in the medium. Next, 2 µL of 2 mM sulfo-NHS-Biotin was added to the suspension and the reaction mixture incubated for 1 h on ice in the dark, vortexing every 15 min. The magnetosomes were washed with 100 µL of cold HEPES buffer to remove the sulfo-NHS-Biotin solution. Magnetosomes were resuspended in 100 µL of cold (4 °C) 20 mM HEPES buffer (pH 7.4) then 5 µL of streptavidin Alexa Fluor^®^ 594 (Thermofisher S11227, Waltham, MA, USA) or Alexa Fluor^®^ 488 (Thermofisher S32354). The reaction was incubated for 1 h on ice in darkness with agitation every 15 min and washed 3 times with 100 µL of 20 mM cold HEPES buffer to remove non-bound streptavidin. 

#### 2.1.4. Synthesis of MNP

A room temperature co-precipitation was used to prepare both cMNP and rMNP. cMNP were synthesized using 0.5 molar ratio of ferric to ferrous iron salts. The mixed iron salts (127 mg of Fe_2_(SO_4_)_3_ and 139 mg of FeSO_4_) were dissolved in 20 mL of N_2_ sparged milliQ. After, 8 mL of 500 mM NaOH was added dropwise at a rate of 50 μL min under constant stirring for 2.45 h until a black precipitation is formed. The MNP were magnetically collected with a neodymium magnet and washed with degassed milliQ water 3 times.

rMNP were synthesized using the same co-precipitation process as described above but reversed: Ferric and ferrous iron salts were dissolved in 8 mL of nitrogen sparged milliQ and added to a 50 mL flask containing 200 mM NaOH solution in milliQ at a rate of 50 µL min under a nitrogen atmosphere.

#### 2.1.5. Surface Coating of MNP with Oleic Acid

cMNP were dispersed by sonication in 200 mL of methanol under nitrogen for 15 min, to which 50 mL of oleic acid was added under stirring at 80 °C. The coated cMNP (OA@cMNP) was filtered and washed with acetone, then the precipitate was dried at room temperature [35].

#### 2.1.6. Surface Coating of MNP with Silica and Fluorescent Silica

Either cMNP or rMNP were dispersed in ethanol and sonicated under a nitrogen atmosphere for 15 min. After sonication, 6 mL of milliQ and 3 mL of 30% ammonium hydroxide (Sigma Aldrich) were added at room temperature under stirring; following this, 400 µL of tetraethyl orthosilicate (TEOS) was added and stirred under nitrogen for 5 h. The silica coated MNP were then washed in milliQ and dried in vacuum over. 

Fluorescent Si@rMNP were also prepared (for detection by flow cytometry) via doping with Rhodamine B isothiocyanate (RITC). Here, 10 mg of Si@rMNP was suspended in 50 mL of Toluene (99.9%) (Sigma Aldrich) by sonication for 15 min. Then, 2 mL of ammonium hydroxide (30%) and 200 µL of RITC conjugated (3-aminopropyl) triethoxysilane (APTES) was added whilst stirring for 1 h at room temperature before being washed in toluene (99.9%) and milliQ and dried in a vacuum oven.

#### 2.1.7. Surface Functionalization of Si@rMNP: EDC-NHS Coupling

A carbodiimide bond was formed through activating 50 µg of epriubicin with 75 µL of EDC (5 mg/mL) and 50 µL of NHS (5 mg/mL) and left to mix for 30 min. After, 100 µg of Si@rMNP were added to the mixture and left to rotate for a further 2 h before being transferred into a cold room (4 °C) overnight. Unbound epirubicin was removed through magnetic washing in PBS.

### 2.2. Methods of Cancer Cell Culture

MDA-MB-231 breast epithelial adenocarcinoma cell lines purchased from ECACC were grown at 37 °C, 5% CO_2_ in Dulbecco’s Modified Eagles Medium (DMEM) containing 4.5 g/L glucose with L-glutamine (A549, HCT116 p53 +/+, HCT116 p53 −/−, LLC-1) supplemented with 10% fetal calf serum and non-essential amino acids (Lonza). Cells were passaged when 70–90% confluent and used with 10 passages of thawing. Cells were tested regularly for mycoplasma and authenticity check

### 2.3. Characterizations

#### 2.3.1. Chemical Characterization

Surface Analysis: Fourier Transform Infrared Spectroscopy (FTIR)

Surface analysis of the functional groups was performed using FTIR (Perkins Elmer, Waltham, MA, USA) with analysis performed using Spectrum 10 with scans from 400–4000 cm^−1^ with baseline correction of all samples. Then, 1.5 mg of sample was ground with 150 mg of potassium bromide and heated to 90 °C before being pressed into a disc for analysis.

##### Crystal Structure: X-ray Diffraction (XRD)

Crystal structure of the nanoparticles was determined by XRD (Bruker AXS advanced, Billerica, MA, USA) on a capillary stage with collection between 20° to 80°. 

##### In Solution Characterization: Dynamic Light Scattering (DLS) and Zeta Potential 

In-solution size was measured by DLS (Malvern Analytics, Worchester, UK) 0.01 mg/mL of particle in milliQ water was measured to determine the hydrodynamic size of the particles. The surface charge of the particles was measured via zeta potential (Malvern Analytics) at pH 7. 

##### UV-Visible Spectroscopy of Epirubicin Conjugates

A Varian Cary 50 UV-Vis spectroscopy (Agilent, Santa Clara, CA, USA) was used to determine the conjugation efficiency of the epirubicin on the MNP. Fifty micrograms of epirubicin and 100 µg functionalized MNP were resuspended in milliQ water and provided the positive and negative controls, respectively. The samples were diluted accordingly and recorded using the Varian Cary 50 software.

##### Elemental Analysis: Inductively Coupled Plasma Optical Emission Spectroscopy (ICP-OES) 

MDA-MB-231 cells (5 × 10^4^ per well) were cultured in a 24-well plate, cells were treated with Si@cMNP at concentrations of 0, 0.022, 0.034, and 0.17 mg/mL, and Si@rMNP at concentrations of 0, 0.005, 0.025, 0.050, and 0.100 mg/mL concentration were administered to the cells and incubated for 24 h. After incubation, the cells were washed in PBS before trypsinization and digestion in aqua regia (approximately 1 mL 3:1 mixture of hydrochloric to nitric acid) and made up to a final volume of 10 mL with water. The iron concentration was determined via ICP-OES (Spectro Green, Spectro, Kleve, Germany) then calculated back to the total iron taken up by the cells in the well.

#### 2.3.2. Microscopy

##### Transmission Electron Microscopy (TEM)

Sample preparation for MNP: 10 µL of the nanoparticle suspensions in water was dispensed onto the surface of a carbon coated copper electron microscopy grid and left for 1 min. After which, the excess liquid was removed via blotting and dried using a vacuum line. 

Sample preparation for intracellular MNP: MDA-MB-231 cells were incubated with 0.2 mg/mL of magnetosomes and cMNP, OA@MNP, and Si@rMNP. After 24 h incubation, the cells were collected and fixed with 3% glutaraldehyde solution, 0.1 M sodium cacodylate, and fixed with 2% osmium tetroxide, before being dehydrated, cleaned in epoxypropane (EPP), and embedded in a raldite resin. Ultrathin sections, approximately 85 nm, were cut (leica UC6 ultramicrotome, LeicaWetzlar, Germany) onto 200 mesh copper grids. The sections were stained with uranyl Acetate followed by Reyonld’s lead citrate.

All grids were examined on a FEI Tecnai G2 Spirit TEM (accelerating voltage 80 kV) (FEI, Lausanne, Switzerland). Nanoparticle size analysis was measured using ImageJ software and the size distribution was calculated in GraphPad Prism.

##### Bright field and fluorescence microscopy of Alexa fluor^®^ 488 modified magnetosomes

0.05 mg/mL of labelled magnetosomes were dosed into cells cultured (3 × 105 MDA-MB-231 cell). Observation of samples processed for bright field and fluorescence microscopy were made with a Zeiss Axio Observer Z1 Motorized Inverted Fluorescence Microscope (Nikon, Minato City, Tokyo, Japan) and processed using the gatan digital micrograph software (Gatan, Pleasaton, CA, USA). 

##### Fluorescence Microscopy of Modified Si@rMNP

MDA-MB-231 Cells were incubated with Rhob Si@rMNP (0.022 mg/mL, 0.035 mg/mL, 0.17 and 0.35 mg/mL) for 24 h. After incubation, cells were washed and fixed with 4% paraformaldehyde (Boster Bio, Pleasaton, CA, USA) 1:1000 dilution of Alexa Fluor 488^®^ phalloidin (Santa Cruz) and 1:1000 dilution of DAPI (Sigma Aldritch). Cells were washed twice in 500 µL of PBS and mounted on to microscope slides by inverting onto Thermo Shandon Immuno Mount (ThermoFisher Scientific, Waltham, MA, USA) and the edges sealed with nail varnish. Imaging of cells was performed using a 60× objective on a Nikon TE200 Inverted Fluorescence and Phase Contrast Microscope (Nikon, Minato City, Tokyo, Japan). Images were taken using separate channels and merged using ImageJ software.

#### 2.3.3. Flow Cytometry

##### Cytotoxicity of Nanoparticles

MDA-MB-231 cells were incubated with magnetosomes, OA@MNP, Si@cMNP, and Si@rMNP at concentrations of 0.022 mg/mL, 0.034 mg/mL, 0.17 mg/mL, and 0.35 mg/mL for 24 h after which they were resuspended with 5 µL of 10 µg ml propidium iodide (PI) for analysis.

##### Uptake of Particles

MDA-MB-231 cells were incubated with magnetosomes, OA@MNP, and cMNP at concentrations of 0.022 mg/mL, 0.034 mg/mL, mg ml, 0.17 mg/mL, and 0.35 mg/mL for 24 h prior to fixing and staining with propidium iodine. Uptake of nanoparticles was then determined via Flow cytometry analysis after 24 h incubation, cells were removed from the wells, and suspended in PBS before being analyzed using a LSRII (BD Biosciences, Franklin Lakes, NJ, USA). Uptake of magnetosomes and OA@MNP were determined via the increased granularity of the cells, Si@rMNP were incubated at concentrations of 0.005, 0.025, 0.05, 0.1, and 0.2 mg/mL and determined via the presence of a fluorescent marker Rhodamine B isothiocyanate due to the small size, and analysis was performed on flowJo software (BS Biosciences, Ashland, OR, USA).

## 3. Results and Discussion

In this study, we compare biological and synthetic methods of producing MNP and assess their uptake into cancer cells. Magnetosomes are the biosynthesized sample, while room temperature co-precipitation is used to chemically synthesize the control MNP (cMNP). These are then coated to form biomimetic MNP. The MNP are coated with oleic acid (OA@cMNP) or with silica (Si@cMNP). A quicker reverse room temperature co-precipitation method is also used to synthesize smaller MNP (rMNP). These and the silica coated versions (Si@rMNP) were assessed to compared the impact of size. Chemically synthesized MNP and magnetosomes both have distinct advantages that they pose as well as drawbacks (Table 1), but do the advantages of increased simplicity and speed of production result in significantly superior MNP compared to magnetosomes?

### 3.1. Synthesis and Characterisation of Magnetosomes, cMNP, OA@cMNP, Si@MNP, rMNP, and Si@MNP

AMB-1 was grown in a microaerobic iron rich environment before being lysed to extract magnetosomes. Magnetic purification was performed to remove any cell debris resulting in chains of magnetosomes being isolated (Figure 2a). Synthetic control particles (cMNP) were synthesized using a room temperature co-precipitation method or a reverse room temperature co-precipitation method (rMNP) (Figure 2b,c). The size of the MNP was greatly affected by the method of synthesis with the rMNP producing much smaller particles than the cMNP, and this is due to there being fewer available iron ions post nucleation in the reverse precipitation method, so more nucleation occurs with less iron available for particle growth. This is because addition of the iron solution to the base in the reverse reaction will always mean a lower concentration of available iron when the MNP forms [36]. While in the control reaction, there is more available iron and it is supplied for longer, as the ferrous species slowly converts to grow larger particles than is seen for precipitations using a precursor solution with a higher ferric ratio [37]. The resulting MNP were then coated with either oleic acid (OA@cMNP) or silica (Si@cMNP/Si@rMNP) (Figure 2d–f, respectively) to mimic the magnetosomes with the aim of leading to improved biocompatibility and to provide a suitable surface for further functionalization. 

#### 3.1.1. Size Analysis

Figure 2 shows TEM images and size analysis of all samples. Figure 2a shows that the magnetosomes (Figure 2a) have a size of 46 ± 11.8 nm (Table 2), which is larger than the cMNP (Figure 2b) of 32 ± 9.9 nm (Table 2). The size of the coated OA@cMNP (Figure 2d) and Si@cMNP (Figure 2e) is 5 61 ± 18.2 nm and 49 ± 12. Nm, respectively (Table 2), and thus very comparable to the size of magnetosomes. The rMNP (Figure 2c) produced the smallest particles with a size of 8 ± 4 nm (Table 2), which increased upon coating with silica (Si@rMNP) (Figure 2f) to 19 ± 11 nm (Table 2). Figure 2g shows the best fit size distribution of the diameters of MNP measured for all samples. Individual histogram data is shown in Appendix A. This thickness of the coatings is clearly shown in Figure 2g with the arrows from uncoated to coated particles. Although the size of all the MNP vary, silica forms a thinner coating of approximately 4–8 nm thick while oleic acid forms a coating approximately 15 nm thick. The oleic acid coating is thicker than expected and is assumed to be a bilayer of oleic acid. It is interesting to note that the size distribution is much broader for the synthetic coated cMNP compared to the magnetosomes with the ranges of the OA@cMNP being almost twice that of the biogenic particles, 71 nm range for Si@cMNP and 98 nm range for OA@cMNP compared with 55 nm for magnetosomes (Figure 2g). This difference suggests that the coating thickness in the synthetic coated cMNP has much less uniformity and control. The rMNP and Si@rMNP samples showed a similar relative distribution range compared to magnetosomes showing the reverse precipitation synthesis to be much more precise. Magnetosomes showed the smallest percentile size distribution compared to the MNP (Figure 2g), which is to be expected due to the strict control during the biomineralization process [38].

#### 3.1.2. XRD

XRD was used to confirm the crystalline structure and size of each of the samples (Figure 2a and Table 2). Diffraction peaks of 2θ at 30.3, 35.6, 43.4, 53.7, 57.3, and 62.9 correspond to the crystal planes [220], [311], [400], [422], [511], and [440] of magnetite, indicating the major component in all MNP samples is magnetite [39,40]. 

Crystallite size was estimated from the [311] peak using the Scherrer equation (Table 2). This showed particle sizes similar to those obtained via TEM. Note that the coating should be invisible to XRD, so sizes for the all the cMNP group and rMNP group should be similar, regardless of coating or lack of coating. Differences between the observed size of uncoated MNP via TEM and the size determined via the Scherrer equation may be due to a number of factors such as a lower number (but larger volume) of larger particles making a smaller impact on TEM measurements but dominating the XRD signal. 

#### 3.1.3. Surface Analysis

The surface charge of all MNP affect how they interact with the surface of cells, and therefore affects uptake and cytotoxicity. A more positive surface charge is associated with greater uptake but a higher level of toxicity. The surface charge of MNP was measured using ζ potential (Table 2). Magnetosomes have a surface charge of −43.79 mv which is comparable to that of OA@MNP (−50.73 mv), showing similarity in the surface coatings. Notiably this is also similar to cMNP (−49.95 mv). Zeta potential is known to be effected by particle size [41]. The increase in zeta potential seen from the larger cMNP (−49.95 mv) to the rMNP (−22.9 mv) can possibly be explained by the increase in proton accumulation along the edge of the smaller particles [42]. The presence of the silica coating increases the charge considerably from −49.95 to −27.3 mv (Si@cMNP) and slightly from −22.9 to −25.4 mv (Si@rMNP), suggesting silica coated MNP will be taken up more readily by cells.

DLS measurements give the hydrodynamic size of the particles as they appear in solution, and so can be affected by factors such as surface charge and agglomeration. Due to there being a range of particle sizes (not monodispersed), this can also lead to an increase in the size observed by DLS due to the increased weighting given to larger particles in DLS. Clearly, the coating is critical in these factors; when a silica coating is added to rMNP, the hydrodynamic size reduces due to the increased dispersity that the coating offers. However, this is not seen when cMNP are coated with silica. This can be explained by aggregation for a different reason. The larger cMNP samples are single domain magnets that are attracted to each other and so will aggregate creating large clusters whereas smaller rMNP are superparamagnetic so will not aggregate to the same degree [43]. This explains why cMNP has such a large hydrodynamic size. A coating will further affect aggregation, as it will prevent the closest and strongest magnetic interactions, so a thicker coating will allow better dispersion in solution. This is clearly seen with magnetosomes and OA@cMNP with a reduced hydrodynamic size compared to cMNP. The silica coating is thinner and as such may not prevent aggregation. Si@cMNP aggregated to such an extent they could not remain in solution long enough to take a measurement.

Fourier transform infrared spectroscopy (FTIR) was performed across the wave number range of 400–4000 cm^−1^ for all samples (Figure 3b). Both the cMNP and the rMNP showed the same profile so only rMNP is shown. The peak at 586 cm^−1^ corresponds to Fe-O vibration bands, and the peaks at 3303 and 1642 cm^−1^ can be attributed to water molecules on the surface of the MNP giving spectra representative of magnetite in the literature [38,39,40]. The spectrum of coated OA@MNP shows five bands appearing at 1432, 1544, 1709, 2853, and 2924 cm^−1^. The absorption band at 1709 cm^−1^ shows the stretching vibration of the C=O of a carboxyl group, which indicates the presence of oleic acid on the cMNP surface [21,44,45,46,47]. The polar carboxylic acid head groups of the oleic acid are coordinated to the surface of magnetite through the oxygen, present in the carboxylate group, to the iron ions [35]. The nonpolar tails, therefore, extend into the solution offering a hydrophobic coating [35]. The last two bands, at 2924 and 2853 cm^−1^, correspond to the symmetric and asymmetric CH_2_ stretching of the oleic acid [39]. FTIR of the magnetosomes show carboxylic acid, amine, amide, and phosphate functional group peaks with vibrations at 3343, 1580, 1634, and 1054 cm^−1^. From the spectrum, it can be seen that the bending vibrations of the primary amino group occur at 3343 and 1580 cm^−1^ [23,48,49]. A peak at 3343 cm^−1^ is indicative of the existence of an OH/NH group present on the magnetosomes. A peak also appears at 1634 cm^−1^ (mainly due to C-O stretching), which suggests the existence of amide I and amide II bonds in the protein-peptide bond, due to absorption of protein [49]. The presence of these peaks after extraction of the magnetosomes suggest that the phospholipid bilayer containing magnetosome associated proteins remains and this is supported by the TEM image of the chain of magnetosomes as no aggregation is present (Figure 2a) [49]. The Si@rMNP spectrum shows the characteristic Fe-O stretching band of magnetite as well as a large band at 1098 cm^−1^ indicating Si-O-Si confirming the presence of silica on the MNP. The Si@rMNP and Si@cMNP showed very similar spectra (Si@cMNP is not shown).

### 3.2. MNP Interactions with MDA-MB-231 Cells

#### 3.2.1. Uptake and Localisation

The effect of MNP size on uptake into MDA-MB-231 cells was studied using Si@cMNP and Si@rMNP as these particles only differ in size. Figure 4 shows ICP OES (detecting elemental iron) analysis of MDA-MB-231 cells after incubating with either Si@cMNP or Si@rMNP to assess particle uptake. The cells were incubated with concentration of particles below 0.2 mg/mL. MNP concentrations of 0.2 mg/mL and above lead to aggregation and difficulty removing residual/surface MNP giving unreliable data with large errors. The data show that particle size in this range has a negligible effect on overall iron uptake by the MDA-MB-231 cell line. The si@rMNP have a slightly higher value at higher concentrations, although it cannot be considered significant. It should be noted that the elemental analysis technique reports quantity of iron, therefore similar or slightly higher quantities of iron for Si@rMNP suggest a slightly smaller number of larger sized Si@cMNP are up taken than smaller Si@rMNP. While the quantity of iron up taken is similar for both, the smaller Si@rMNP are likely to offer a better option for delivery to cells due to the increased number of particles, larger surface area, less propensity to aggregate, and their smaller hydrodynamic size allowing them better mobility around the body (hydrodynamic size > 200 nm results in accumulation within the liver and spleen) [50]. For this reason, out of the two, only Si@rMNP are taken forward for further cell uptake analysis. 

Figure 5a–d shows TEM images of MDA-MB-231 cells after uptake of magnetosomes, cMNP, Si@rMNP, and OA@cMNP, respectively and all show the MNP were readily taken up within the cell through inclusion bodies and were located intracellularly. Due to the size of the particles, intracellular uptake is most likely via pinocytosis with the inclusion bodies being pinosomes or lysosomes [51]. Although other processes have been proposed for MNP internalization including clathrin mediated endocytosis, caveolin mediated endocytosis and direct penetration of the membrane [52,53]. Although still present in some of the intracellular vesicles, a large proportion of particles are seen to have migrated to the extracellular matrix. The majority of the magnetosomes were presumed to be still organized in chains, which may help them to be less prone to uncontrolled aggregation. Indeed, several chains of magnetosomes appear inside the cell (Figure 5a). When magnetosomes are distributed in a small stable chain, they are less likely to aggregate than other MNP [54].

The ability to be taken up by cancer cells offers the ability to allow delivery of therapeutic compounds directly into the cell and the use of their magnetism to steer the MNP within the body. This would allow therapeutics to be delivered directly to the target site [12,55]. 

The amount of uptake of MNP and magnetosomes was compared via flow cytometry to measure the percentage of cells that have taken up particles, increased granularity of cells was used to determine uptake in magnetosome, cMNP, and OA@cMNP but due to the smaller diameter of the Si@rMNP a fluorescent tag was used to identify % of cells that had taken up these smaller particles (Figure 5e). There is a similar trend observed in all concentrations tested showing an increase in the amount of MNP uptake as the concentration of MNPs increased. There is some variation in the amount of particles taken up, and this may be an effect of surface properties of each particle [56].

#### 3.2.2. Cellular Viability

The toxicity and tolerance of MNP are crucial factors in reducing side effects of MNP-based therapies. Figure 6 shows MDA-MB-231 cell survival over exposure to increasing MNP concentrations. Magnetosomes showed a decrease in cell viability as the concentration of particles increased when compared with the control, suggesting some toxicity (Figure 6). Although the highest concentration tested showed no significant difference, magnetosomes are the only particle that showed consistent effect on cell viability across a range of concentrations. cMNP were shown to effect cell viability at a concentration of 0.18 mg/mL and the OA@cMNP were shown to negatively affect cell viability at a concentration of 0.35 mg/mL. The effect of the OA@cMNP at the concentration of 0.35 mg/mL may indicate an increased negative effect on cell viability at concentrations higher than 0.35 mg/mL.

When comparing particles at each concentration there is seen to be a significant difference (*p* < 0.05) in only the Si@rMNP and magnetosomes at the highest concentration of 0.35 mg/mL. This suggests that even though there is a decrease in cell viability in the magnetosomes compared with the control, there is very little difference between the tested MNP, indicating that there is only a small negative effect on cell viability at the concentrations tested.

### 3.3. Functionalization

The ability to bind compounds to the surface of magnetosomes or the most promising magnetosome mimics allows for further development of them as viable cancer therapeutics. It would allow therapeutic compounds to be attached directly to the magnetosomes or Si@rMNP and therefore increase treatment efficiency through targeting of specific cells and steering to the site of interest.

#### 3.3.1. Biotinylation of Magnetosomes

Biotin streptavidin binding is a fast and specific method to functionalize NPs. Biotinylation is an ideal conjugation method for use with magnetosomes due to its specificity and ability to form covalent bonds between the biotin molecule and proteins on the surface of the magnetosome. There are two methods of biotinylation: Chemical and enzymatic. Here we used the chemical method owing to its simplicity. Chemical biotinylation was performed by modifying the surface amino acids of the magnetosome with biotin N-hydroxysuccinimide (NHS). This was used to bind a green or red fluorescence Alexa Fluor^®^488 or Alexa Fluor^®^ 594 streptavidin to magnetosomes. The functionalized magnetosomes were imaged using fluorescent microscopy to determine if the Alexa Fluor^®^ 594 and 488 dye was bound to the magnetosomes surface (Figure 7a–i). From microscopy, the streptavidin Alexa Fluor^®^ 594 dye is seen to only bind to the magnetosomes after biotinylation (Figure 7a–f) as unbiotinylated magnetsomes have little to no fluorescence after treatment with the dye (Figure 7a–c). When biotinylated, the 594 and 488 dye is localized on the magnetosomes after washing (Figure 7c,f), showing conjugation to the magnetosome membrane. Incubation of MDA-MB-231 cells with the conjugate were imaged after 24-h (Figure 7j–l) and showed uptake of the magnetosomes into cells (Figure 7j) as well as fluorescence of the magnetosomes from the conjugated Alexa Fluor^®^488. The transport and internalization of these biotin functionalised magnetosomes was observed inside the cells (Figure 7l). The magnetosomes can be seen within the cytoplasm of the cell as well as a portion being closely associated with the nuclear membrane.

#### 3.3.2. EDC-NHS Coupling to Si@rMNP

The use of EDC-NHS coupling allows for more targeted binding to the functionalized surface of the Si@rMNP, as this method reduces the risk of cross linking of the NH_2_ functional groups on the surface of the Si@rMNP, which could result in particle–particle binding as well as binding to the target compound. This particle–particle binding could result in the formation of large clusters of particles. By using EDC-NHS binding targeting the carboxylic acid functional groups (COOH) found on epirubicin and crosslinking them with the amine functionalized surface of the Si@rMNP, this can avoid the risk of particle-particle bind by having a more targeted approach.

From the results, it can be clearly seen that epirubicin is bound to the surface of the Si@rMNP (Figure 8a–c) with the fluorescent epirubicin showing localization to the surface of the Si@rMNP even after extensive washing (Figure 8c). This is supported by the UV-vis analysis showing the presence of epirubicin characteristic peaks at 230 and 250 nm wavelengths, which match the reference spectra of epirubicin [57]. In bound Si@rMNP, even after extensive washing, the amount of epirubicin present is dependent on the amount in solution during coupling, with higher concentrations showing increased amounts of epirubicin bound to the Si@rMNP; therefore, the binding of epirubicin to the Si@rMNP is a concentration-dependent process (Figure 8d). Epirubicin binding to magneite (50 µg of epirubicin to 100 µg of Si@MNP) was seen to not hinder the uptake of the Si@rMNP as the particles were clearly seen localized with the MDA-MB-231 cells with epirubicin (Figure 8e). This showed that the epirubicin did not inhibit the uptake of the Si@rMNP, but it is unclear from this whether the epirubicin still functions as intended.

These results show that surface modified magnetosomes and Si@rMNP are both taken up by MDA-MB-231 cells and that the presence of the biotin/fluorescent streptavidin or epirubicin does not seem to hinder cell uptake. Alexa Fluor^®^488 streptavidin conjugated magnetosomes and epirubicin Si@rMNP are taken up readily into the cell (Figure 7f and Figure 8e, respectively). This allows the delivery of a molecule directly within the cell the requirement to release the payload (the magnetosome and the Alexa Fluor^®^488 streptavidin separating) showing an ability to be reliable and able to deliver compounds directly to within cells.

## 4. Conclusions

Although there has been development in the use of MNP for diagnostics and in hyperthermia treatments, there is less development in the use of MNP for drug delivery or comparison between MNP for cell uptake and toxicty. In this study, we assessed magnetosomes against coated synthetic MNP as magnetosome mimics. OA@cMNP and Si@cMNP were the closest in size to magnetosomes. OA@cMNP also had a similar surface charge and slightly larger hydrodynamic size compared to magnetosomes whereas Si@cMNP had a less negative surface but were found to aggregate and fall out of solution. Interestingly, a biomimic that was smaller (Si@rMNP) was able to better disperse in solution and had a less negative surface charge that is known to favor uptake. There was no significant difference between the uptake of silica coated MNP of either size by cancer cells. This is by mass of iron, so more smaller particles are up taken in the Si@rMNP sample. All synthetic biomimetic coated MNP showed no toxicity to MDA-MB-231 cells, while the magnetosomes showed a slight toxicity with increased concentration. Interestingly, the magnetosomes showed the lowest uptake by MDA-MB-231 cells while the Si@rMNP showed the highest. It is well known that magnetosomes have enhanced therapeutic properties so magnetosomes along with the most promising biomimic (Si@rMNP) were taken forward to demonstrate functionalization. Biotinylation of magnetosomes is an easy way to functionalize magnetosomes, allowing conjugation to any streptavidin modified molecules. The biotinylation does not appear to prevent uptake of the magnetosomes into the cell. By using biotinylated magnetosomes, we successfully conjugated a fluorescent marker showing the possibility of attachment of chemotherapeutics or targeting moieties for an enhanced therapeutic benefit. Similarly, EDC-NHS coupling of epirubicin to the amine functionalised Si@rMNP surface was simply performed and did not block cellular uptake with the epirubicin Si@rMNP seen intracellularly.

This demonstrates that both magnetosomes and magnetosomes mimics can be successfully functionalized and taken up by cancer cells. Further development of the use of MNP to target cancer and other diseases will show their promise as a targeted therapeutic delivery vehicle as well as hyperthermic and photothermic therapies, and functionality delivers further theragnostic potential. While further studies are needed to assess the therapeutic efficacy of these in cancer models compared to more traditional nanoparticle vehicles, as well as to explore different linkers and drug conjugations, this study lays the foundations to develop this approach for magnetic nanomedicine.

## Figures and Tables

**Figure 1 pharmaceutics-13-00367-f001:**
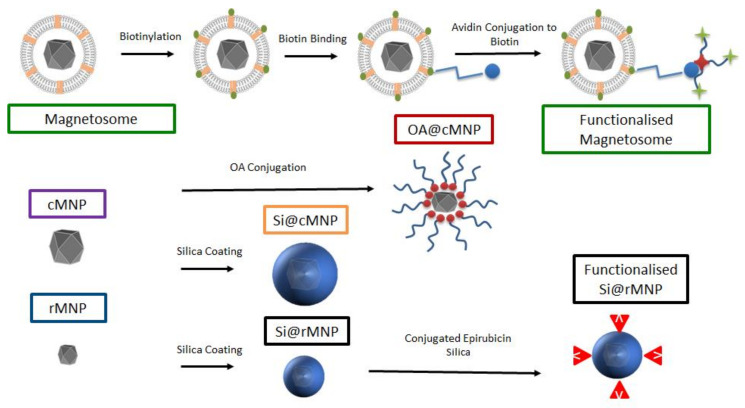
A schematic of the experimental design and samples produced in the study. Top labelled in green depicts the magnetosome and how it is biotinylated for functionalization. Centre left shows the control cMNP labelled in purple with surface coatings of oleic acid (red) and silica (orange label). Bottom left shows smaller control rMNP (blue label) coated with silica (black label) and with conjugated epirubincin (bottom right). Sample color-code used throughout in figures.

**Figure 2 pharmaceutics-13-00367-f002:**
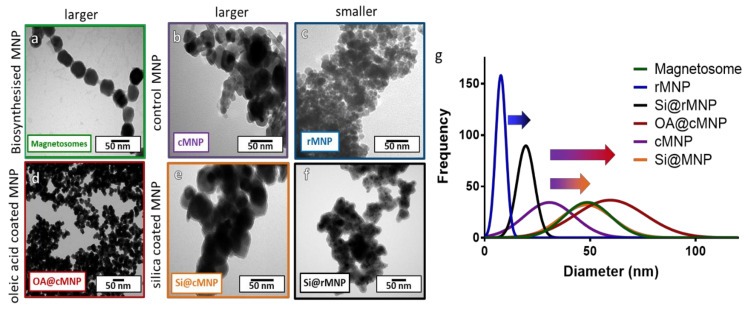
TEM images of magnetosomes and synthesized (un)coated MNP with size distribution. (**a**) Chains of magnetosomes, (**b**) cMNP, (**c**) rMNP, (**d**) OA@cMNP, (**e**) Si@cMNP, (**f**) Si@rMNP, (**g**) frequency distribution of magnetosomes and MNP sizes (curves show fitting curves of original histograms presented in the Appendix A).

**Figure 3 pharmaceutics-13-00367-f003:**
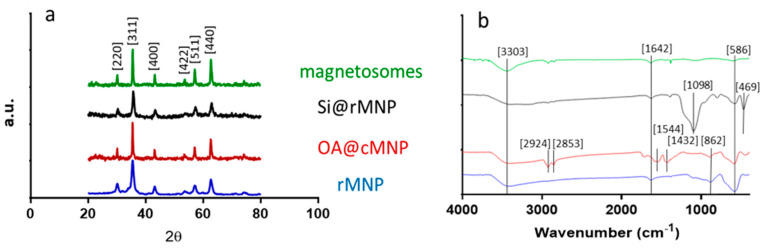
Structural and chemical analysis of the magnetosomes and synthesized MNP. (**a**) XRD spectra showing the diffraction peaks with 2θ angle measurements from 20 to 80. (**b**) FTIR spectra with wavenumber from 400–4000 (cm^−1^).

**Figure 4 pharmaceutics-13-00367-f004:**
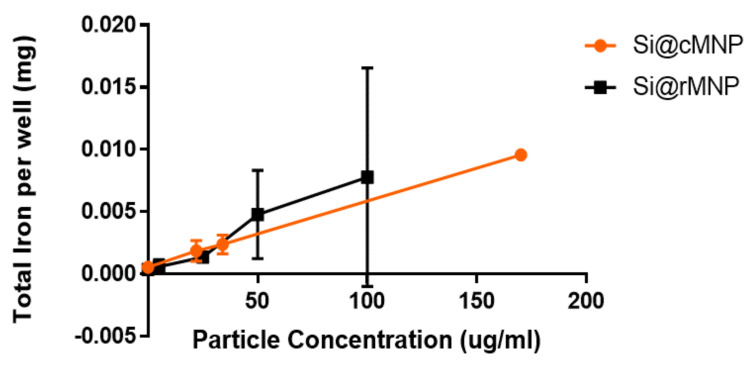
Iron uptake into cells of Si@cMNP (orange) and Si@rMNP (black). ICP OES showing the amount of iron present within 50,000 MDA-MB-231 cells after 24 h incubation with 0, 0.022, 0.034, and 0.17 mg/mL of Si@cMNP, and 0, 0.005, 0.025, 0.050, 0.100 mg/mL of Si@rMNP.

**Figure 5 pharmaceutics-13-00367-f005:**
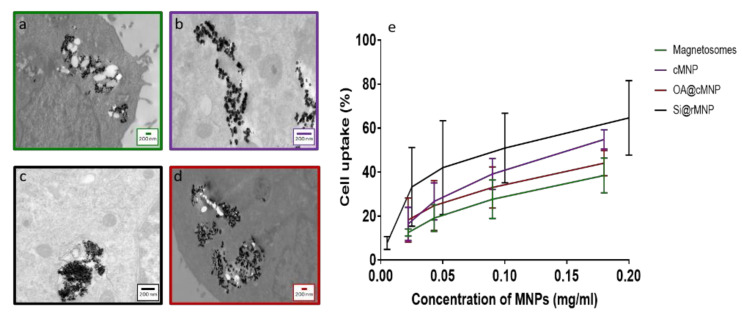
Intracellular uptake of magnetosomes and MNP. TEM slices of MDA-MB-231 cells after 24 h incubation (**a**) Magnetosomes, (**b**) cMNP, (**c**) Si@rMNP, (**d**) OA@cMNP, (**e**) uptake of magnetosomes and MNP via flow cytometry showing the percentage of cells with particles present magnetosomes, cMNP, Si@rMNP, and OA@MNP at concentrations of 0.022, 0.034, 0.17, and 0.35 mg/mL gated by granularity Si@rMNP were treated at concentrations of 0.005, 0.025, 0.05, 0.1, and 0.2 mg/mL and determined via the presence of a fluorescent marker Rhodamine B isothiocyanate.

**Figure 6 pharmaceutics-13-00367-f006:**
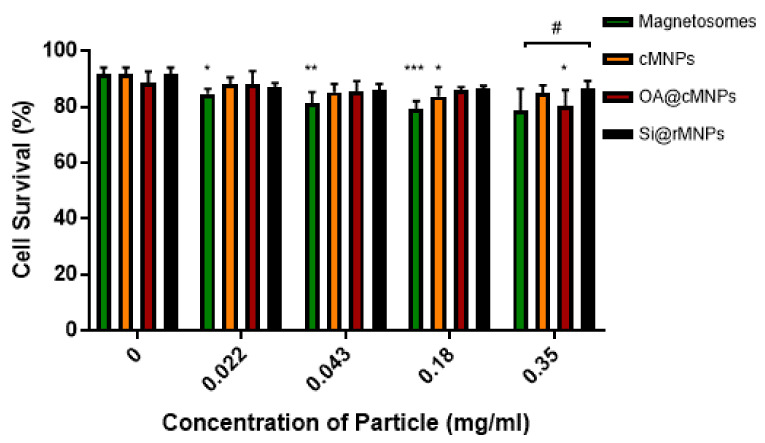
Cell viability as determined by flow cytometry 24 h after treatment with magnetosomes (green), Si@rMNP (black), Si@cMNP (orange), and OA@MNP (red) at increasing concentrations. * *p* < 0.05, ** *p* < 0.01, *** *p* < 0.001 compared to the control for each group, # *p* < 0.05 compared particle concentrations to the other particles within that same concentration by ANOVA.

**Figure 7 pharmaceutics-13-00367-f007:**
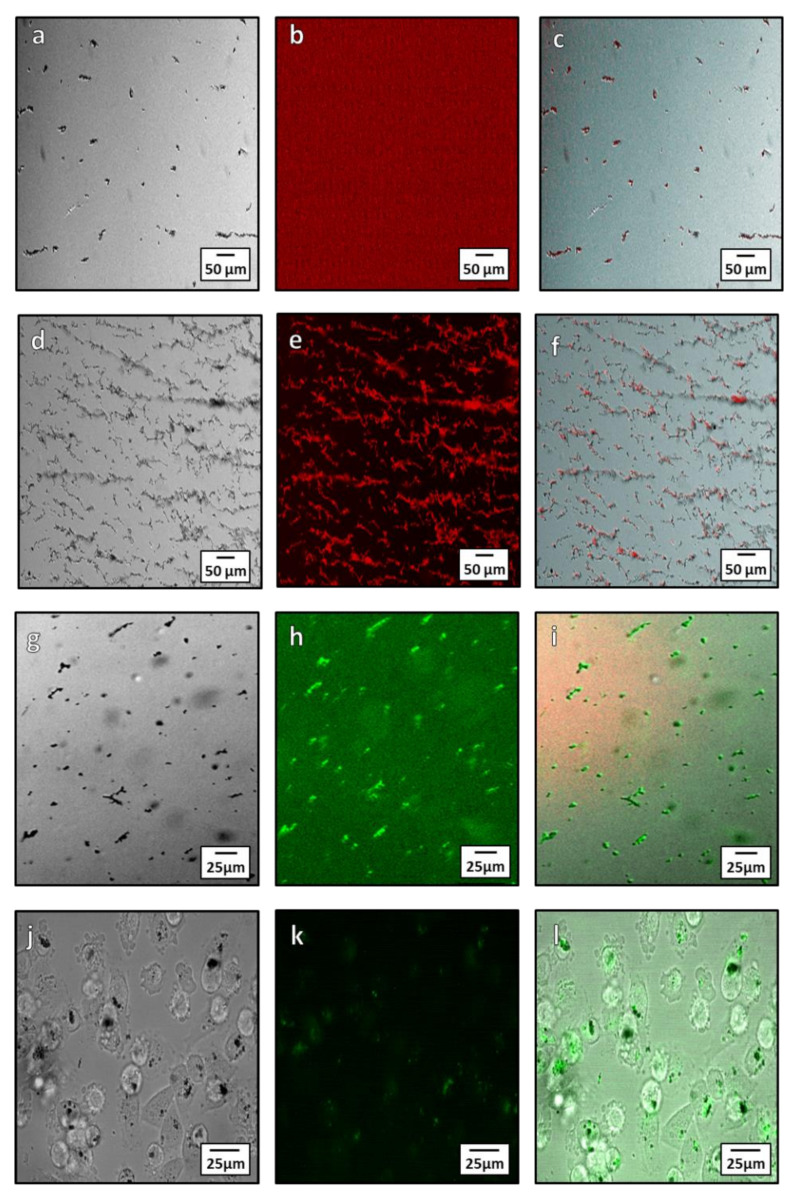
Microscopy showing localization Alexa Fluor^®^ 594 and 488 with magnetosomes after conjugation both intra and extracellularly. (**a**–**c**) unbiotinylated (control with no coupling) magnetosomes treated with streptavidin Alexa Fluor^®^594: (**a**) Brightfield image of unbiotinylated magnetosomes; (**b**) red channel fluorescence showing lack of Alexa Fluor^®^594 fluorescence; (**c**) composite image showing little to no fluorescence bound to the magnetite. (**d**–**f**) Biotinylated magnetosomes conjugated with Alexa Fluor^®^594: (**d**) bright field image of biotinylated magnetosomes; (**e**) red channel of Alexa Fluor^®^594; (**f**) composite showing localization of the Alexa Fluor^®^594 dye to the magnetosomes. (**g**–**i**) Microscopy of Alexa Fluor^®^488 streptavidin conjugation to magnetosomes: (**g**) Bright field microscopy of magnetosome; (**h**) green channel fluorescence showing the Alexa Fluor^®^488 streptavidin; (**i**) composite showing conjugation of the Alexa Fluor^®^488 streptavidin to the magnetosomes. (**j**–**l**) Intracellular Alexa Fluor^®^488 conjugated magnetosome: (**j**) In bright field microscopy; (**k**) fluorescent images of conjugated magnetosomes; and (**l**) a composite of the bright field and fluorescent channels.

**Figure 8 pharmaceutics-13-00367-f008:**
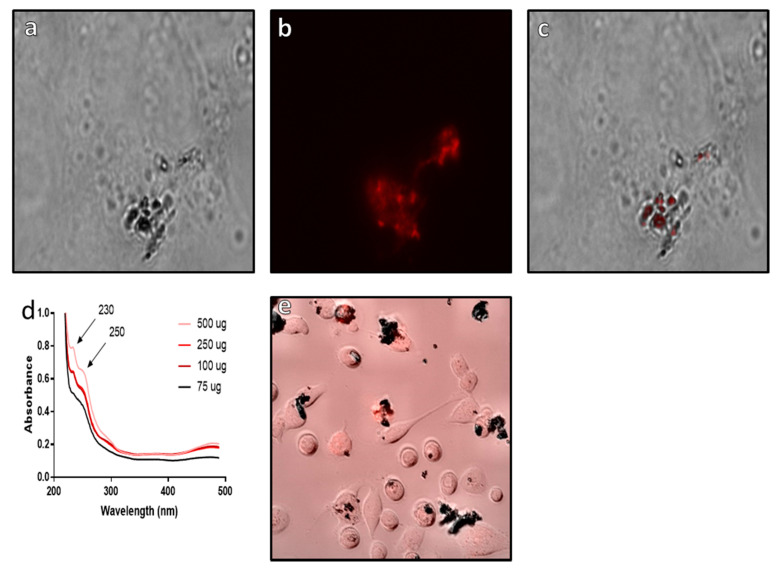
Analysis of epirubicin binding to Si@rMNP. Epirubicin conjugated Si@rMNP under brightfield microscopy (**a**) and fluorescent microscopy (**b**) (images were taken at ×50 magnification), (**c**) a composite image of both bright field and fluorescent channels showing localization of epirubicin to the Si@rMNP, (**d**) UV-vis of Si@rMNP after binding to epirubicin at varying concentrations, (**e**) composite bright field and fluorescent microscopy of epirubicin Si@rMNP uptake with MDA-MB-231 cells.

**Table 1 pharmaceutics-13-00367-t001:** Advantages and disadvantages of both naturally occurring magnetosomes and chemically synthesized magnetic nanoparticles (MNP).

	Magnetosomes		Chemical Synthesis	
	Advantages	Disadvantages	Advantages	Disadvantages
Synthesis	Environmentally friendly, scalable	Slow and Time Consuming, requires specialist equipment	Simple and straight forward, easily scaled	Size control requires stringent experimental parameters.
Surface coating	Synthesized within a lipid membrane	Would need to be extracted from the membrane and recoated	Can be chosen depending on the required function	Further modification if required

**Table 2 pharmaceutics-13-00367-t002:** Characterization data for all MNP, sizing data for TEM obtained from gaussian fit (Figure 3a) of histograms for each sample with standard deviation (Appendix A). Dynamic light scattering (DLS) data are shown in Appendix A.

MNP	Size (nm) (TEM)	Size (nm) (XRD)	Hydrodynamic Size (nm) DLS	Zeta Potential (mv) pH 7
Magnetosomes	46 ± 11.8	51	648.0	−43.79
cMNP	32 ± 9.9	50	1033.5	−49.95
Si@cMNP	49 ± 12.5	40	-	−27.3
OA@cMNP	61 ± 18.2	52	916.7	−50.73
rMNP	8 ± 1.6	12	120.3	−22.9
Si@rMNP	19 ± 3.2	17	81.3	−25.4

## Data Availability

Not applicable.

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
