# Peer review of "Magnetosomes and Magnetosome Mimics: Preparation, Cancer Cell Uptake and Functionalization for Future Cancer Therapies"

_pharmaceutics, 2021, doi:10.3390/pharmaceutics13030367_

Round 1

Reviewer 1 Report

In this paper, the authors investigate some physicochemical properties of magnetosomes and synthetic MNP coated either with silica or oleic acid, their functionalization with fluorescent markers, and uptake by a breast cancer cell line. The topic is of great interest, in general, the investigation methods suit the objectives.

The authors compared the characteristics of the magnetotactic bacteria synthesized magnetosomes with those of two classes of biomimetic magnetic nanoparticles (MNP) obtained by the classical coprecipitation method and further coated with either a silica shell or oleic acid.

The authors obtained some interesting results:

-successful biosynthesis of magnetosomes and their subsequent functionalization with a fluorophore by biotinylation,

-coating of synthetic MNP with either a silica layer or oleic acid and their functionalization with a therapeutic drug (epirubicin),

- assessment of the nanoparticles uptake by the cancer cell lines by different techniques: TEM, ICP-AES, flow cytometry, and their cytotoxicity.

However, there are several issues I suggest the author should address before considering publication:

Formal observations:

  • as a general observation, the resolution of figures is very low, e.g. figure 3a lacks the labels for the diffraction peaks, figure 4 no error bar for the orange curve; figure 5; no scale bar (one can consider that image c has a different magnification as compared to images a, b and d); figure 6 no orange bar (should be purple?); figure 8d the spectrum of 100 uL is indistinguishable  from the one at 250 uL  (I suggest to change the colors) and lack of scale bars for the microscopic images;
  • the units on the ordinate axis in figure 4 should be concentration units, not mass units
  • legend of figure 3 b, the FTIR spectrum limit is 400 cm-1, not 500 cm-1

Other observations:

1.It is not clear how the oleic acid coated MNP can be dispersed in an aqueous environment. The authors state that the OA attaches to the MNP through their polar head, meaning that hydrophobic tails are oriented outwards, rendering the coated MNP hydrophobic. Usually, when MNPs are synthesized in a hydrophobic environment there are specific methods to make them hydrophilic (eg. .treatment with sodium periodate, etc.)

  1. The size analysis leads to some open questions which are not clearly explained in the paper :

-how the coating with oleic acid leads to such a dramatic (almost double) increase in the size (cMNP  32 nm, OA@cMNP 61 nm) as the organic coating is almost transparent for the electrons (could the authors provide a higher magnification image of figure 2 d?), while the XRD sizes are almost similar (50-52 nm?); could it be an artifact due to the aggregation of the hydrophobic MNPs?

-the sizes revealed by the XRD data correspond with the smallest crystallites which are responsible for the largest widths in the XRD peaks; could the authors explain why the XRD sizes are larger than the TEM ones for magnetosomes, cMNP, and rMNP?

- why the silica coating reduces the XRD size for cMNP but increases it for rMNP?

  1. The -23 mV zeta potential of rMNPs is much smaller as compared to the value of -49 mV for cMNPs while the precursors are the same
  2. The FTIR analysis

-the peak at 1642 cm-1 cannot be attributed to the OH stretching as stated in the manuscript

- for the OA@MNPs the peaks at 2853, and 2924 cm-1 do not correspond to the asymmetric and symmetric stretching vibrations of a carboxyl group as stated initially in the manuscript but to the symmetric and asymmetric CH2 stretching of the oleic acid as correctly indicated later  but at slightly different wavenumbers (2927 and 2860 cm-1, which are different from those depicted in figure 3b)

  1. The authors claim that the uptake by the cells is similar for both Si@cMNPs and Si@rMNPS. However, from figure 3 it seems that the uptake of Si@rMNPs is significantly higher. For the Si@cMNPs lacks the highest concentration of 0.2 mg/mL and the error bars. It is not clear why the authors preferred the flow cytometry method (with different labels) to measure the cell uptake instead of ICP AES

  1. I suggest the authors indicate the UV absorption peak wavelengths of epirubicin they referred to in the text and eventually to label them on the spectra

There are some minor spelling errors, missing capitalization. I suggest the authors revise the paper by paying more attention to the details.

Author Response

We thank the reviewer for their time and support for the research and the paper. We are pleased the reviewer saw the interest in the results. We thank the reviewer for their suggestions to improve the manuscript. We have taken these onboard and corrected as outlined below (responses in bold):

Formal observations:

  • as a general observation, the resolution of figures is very low,

Where possible, the resolution of the figures has been improved (in some cases we have swapped for better resolution figures) and clearer labelling has been added (also addressed in comments below).

  • g. figure 3a lacks the labels for the diffraction peaks,

Diffraction peaks have now been added in figure 3a and additionally the peaks labels in figure 3b have been improved so are clearer to read.

  • figure 4 no error bar for the orange curve;

Error bars were present but were very small, so not visible on the graph of that scale and with the symbol of that size. The graph has been redraw (see point 5 below). This larger scale now means the error bars for the Si@cMNP data are now visible with the exception of the highest concentration, which just has a very small difference between the triplicate data readings (thus small error).

  • figure 5; no scale bar (one can consider that image c has a different magnification as compared to images a, b and d);

The Inserts images have been removed and used as the main TEM image as these are clearer and of better quality. Scale bars have now been added. Furthermore the colour-coding of the samples has been made clearer with a key in 5f (as well as stating in the figure caption).

  • figure 6 no orange bar (should be purple?);

We thank the reviewer for highlighting this processing error. It has been corrected. Furthermore, the colour-coding of the samples has been made clearer with a key (as well as stating in the figure caption).

  • figure 8d the spectrum of 100 uL is indistinguishable  from the one at 250 uL  (I suggest to change the colors) and lack of scale bars for the microscopic images;

This is an excellent suggestion, which we have actioned. Note the 250 ad 100 uL are almost overlapping, but hopefully the new more contrasting colours have made this visible. The figure caption has also been improved to describe this figure more clearly.

  • the units on the ordinate axis in figure 4 should be concentration units, not mass units

As the samples are not of a set volume but each from a well of 50,000 cells (then made up to the same 10 ml volume, see methods). The quantity is then calculated back to find the total from the well. Thus the units should be total mg of iron per 50,000 cells. To address this point we have labelled the axis total iron per well (mg) and explained in the figure caption each well contained 50,000 cells.

  • legend of figure 3 b, the FTIR spectrum limit is 400 cm-1, not 500 cm-1

This has been corrected.

Other observations:

1.It is not clear how the oleic acid coated MNP can be dispersed in an aqueous environment. The authors state that the OA attaches to the MNP through their polar head, meaning that hydrophobic tails are oriented outwards, rendering the coated MNP hydrophobic. Usually, when MNPs are synthesized in a hydrophobic environment there are specific methods to make them hydrophilic (eg. .treatment with sodium periodate, etc.)

This is a valid point. Also noted in the paper, this coating is thicker than expected. We assume this is due to multi-layered coating. A double layer will expose the hydrophilic head group, making it soluble. We have added the following sentence to the text. “The oleic acid coating is thicker than expected and is assumed to be a bilayer of oleic acid.”

  1. The size analysis leads to some open questions which are not clearly explained in the paper :

-how the coating with oleic acid leads to such a dramatic (almost double) increase in the size (cMNP  32 nm, OA@cMNP 61 nm) as the organic coating is almost transparent for the electrons (could the authors provide a higher magnification image of figure 2 d?),

While the coatings are more translucent, they are not transparent, and all measurements were performed including the coating. This is indeed a big difference and it is commented on as such in the paper. As said in the previous point, we believe this thickness perhaps explains the ability to disperse.  Here is a section of figure 2d

It can be seen that smaller particles lie in a more translucent organic coating.

 while the XRD sizes are almost similar (50-52 nm); could it be an artefact due to the aggregation of the hydrophobic MNPs? This is possible aggregation during drying of the particles for TEM is always present

The XRD size relates to individual crystallites and this will not be affected by aggregation.

-the sizes revealed by the XRD data correspond with the smallest crystallites which are responsible for the largest widths in the XRD peaks; could the authors explain why the XRD sizes are larger than the TEM ones for magnetosomes, cMNP, and rMNP?

One must remember that this XRD size is obtained from the Scherrer equation and is an estimate based on many assumptions / parameters which may lead to some inaccuracies. The larger size will be due to the parameters and assumptions not ideally fitting with the MNP populations here. The XRD data should give approximately the same values for the top 4 samples in table 2 and the same smaller size for the bottom 2, as XRD is only related to the crystal size rather than the coating. Broadly speaking this is achieved within the limits of this crude method. The discrepancies are probably due to different particle distribution for each sample (if one considers the distribution in supplementary figure S1).

- why does the silica coating reduces the XRD size for cMNP but increases it for rMNP?

As stating in the response above, we do not believe the coating reduces size. This can be seen by comparing figure 2b to e and figure c to f. We believe there is inaccuracy with the calculated crystal size using the Scherrer equation. This is only one way of estimating the particle size. All the data should be considered together.

  1. The -23 mV zeta potential of rMNPs is much smaller as compared to the value of -49 mV for cMNPs while the precursors are the same

We also found this data curious and after several repeats to confirm the values are correct we looked to the literature. This is a known phenomenon due to increasing charge and Brownian motion. The following text and reference have now been added to the manuscript. “Zeta potential is known to be effected by particle size [41]. The increase in zeta potential seen from the larger cMNP (-49.95) to the rMNP (-22.9) can possibly be explained due to the increase in proton accumulation along the edge of the smaller particles [42]. “

  1. The FTIR analysis

-the peak at 1642 cm-1 cannot be attributed to the OH stretching as stated in the manuscript

- for the OA@MNPs the peaks at 2853, and 2924 cm-1 do not correspond to the asymmetric and symmetric stretching vibrations of a carboxyl group as stated initially in the manuscript but to the symmetric and asymmetric CH2 stretching of the oleic acid as correctly indicated later  but at slightly different wavenumbers (2927 and 2860 cm-1, which are different from those depicted in figure 3b)

This is indeed the case and our original text was incorrect. The text has now been corrected with the following:  “The spectrum of coated OA@MNP shows five bands appearing at 1432, 1544, 1709, 2853, and 2924 cm-1. The absorption band at 1709 cm-1 shows the stretching vibration of the C=O of a carboxyl group, which indicates the presence of oleic acid on the cMNP surface [21,44–47]. The polar carboxylic acid head groups of the oleic acid are coordinated to the surface of magnetite through the oxygen, present in the carboxylate group, to the iron ions [48]. The nonpolar tails, therefore, extend into the solution offering a hydrophobic coating [35]. The last two bands, at 2924 and 2853 cm-1, correspond to the symmetric and asymmetric CH2 stretching of the oleic acid [39].”

  1. The authors claim that the uptake by the cells is similar for both Si@cMNPs and Si@rMNPs. However, from figure 4 it seems that the uptake of Si@rMNPs is significantly higher. For the Si@cMNPs lacks the highest concentration of 0.2 mg/mL and the error bars.

It is clear from looking at the raw data that the 0.2 mg/ml data point for Si@rMNP was unreliable data. It had such huge error bars due to particle aggregation in the well, making washing very unreliable, rendering this data point to be of poor quality. After careful consideration and reassessment of the data, we have removed this data point and made reference to this concentration limit on reliability in the text “The cells were incubated with concentration of particles below 0.2 mg/ml. MNP concentrations of 0.2 mg/ml and above lead to aggregation and difficulty removing residual/surface MNPs giving unreliable data with larger errors.”

We also state that our observations are only for this concentration range. Furthermore we address the point by the added comments that “while the si@rMNP have a slightly higher value at higher concentrations, it cannot be considered significant.”

  1. I suggest the authors indicate the UV absorption peak wavelengths of epirubicin they referred to in the text and eventually to label them on the spectra

Reference to the peaks have now been made in the text and arrows indicating the peaks on the spectrum have now been added.

There are some minor spelling errors, missing capitalization. I suggest the authors revise the paper by paying more attention to the details.

The paper has been proof-read carefully by all the co-Authors.

Reviewer 2 Report

The authors characterize and compare magnetosomes, synthetic MNP of different sizes and types of biomimetic MNP for their uptake by cells. The manuscript presents interesting results, but several points need to be clarified:

  1. The synthesized magnetic nanoparticle may comprise several minerals of iron. Magnetic and X-ray diffraction measurements cannot be used to determine the type of mineral that is formed during the synthesis. The authors should use other techniques, for example, Mössbauer spectroscopy or magneto-optical methods.
  2. Have the authors tried to obtain magnetosomes of different sizes? The paper (doi: 10.1109/TMAG.2012.2224098) shows that it is possible to influence the size of the magnetosomes.
  3. Will the use of an alternating magnetic field (magnetic heating) increase the release of binding medicine?

Overall, the presentation is sloppy. The article contains many errors. For example:

  1. Table 1 does not present Zeta potential data.
  2. Fig. 7 description is incorrect. What does panel f shown in Figure 7?
  3. Where is Fig. 9e?

Author Response

We thank the reviewer for their time and support for the research. We thank the reviewer for their suggestions and finding errors for us to correct to improve the manuscript. We corrected as outlined below (responses in bold):

  1. The synthesized magnetic nanoparticle may comprise several minerals of iron. Magnetic and X-ray diffraction measurements cannot be used to determine the type of mineral that is formed during the synthesis. The authors should use other techniques, for example, Mössbauer spectroscopy or magneto-optical methods.

Unfortunately these alternative techniques are not available to us. Although XRD is not definitive proof of magnetite, it is a strong indicator with the peaks that correspond to magnetite being present without the peaks that correspond to similar minerals that could be formed during synthesis so the most likely conclusion is that magnetite has been formed.

  1. Have the authors tried to obtain magnetosomes of different sizes? The paper (doi: 10.1109/TMAG.2012.2224098) shows that it is possible to influence the size of the magnetosomes.

The paper the reviewer mentions is fascinating. While the authors know magnetosomes come in a range of shapes and sizes, it is notoriously difficult to culture many strains. In fact, the only strains the authors culture in their lab are all magnetospirillium and they all produce cubo-octahedral particle in a size range of 45-55 nm.  While it is a fascinating idea to explore if magnetosomes of different size and shape have different cancer cell uptake, from what we have found in our paper, we would predict subtle difference in size will not affect the uptake, but shape may. This could be an interesting suggestion for other labs who are well versed in growing a range of different MTB strains to study, but goes well beyond the scope of this study.

  1. Will the use of an alternating magnetic field (magnetic heating) increase the release of binding medicine?

This is a fascinating question. Indeed there is definitely an excellent follow-on study that could be performed to assess how heat affect different drug binding systems. One could tailor a functionalisation system so the linker is specifically tuned to degrade a certain temperatures. While this is an exciting area of research it is beyond the scope of this study.

Overall, the presentation is sloppy. The article contains many errors. For example:

  1. Table 1 does not present Zeta potential data.

This was a typo, it should have read table “2”. It has been corrected

  1. Fig. 7 description is incorrect. What does panel f shown in Figure 7?

All the figure captions have been reviewed and where required improved. The figure caption for figure 7 has been corrected.

  1. Where is Fig. 9e?

This is a relic from when the paper had a different figure layout, that was not picked up in the proof read. We thank the reviewer for spotting it. It has now been corrected.

Reviewer 3 Report

Authors did not define the type of paper. I suppose it is an Article.

An abbreviation list could be a good way to summarize all the acronyms indicated in this paper.

In the abstract, I would suggest to add the standard deviations in the definition of the mean size of nanoparticles, in the abstract section. In this way, a more precise definition of the produced samples can be given to readers.

The introduction could be enlarged and improved. For example, a brief paragraph containing description and comparison of drug carriers among them could be added (liposomes, magnetosomes, cubosomes, niosomes etc.)

In the introduction, a brief general comment on the external stimuli drug delivery systems could be added, with the proper references.

The total absence of numbered lines made it difficult to provide a proper and fast peer review process.

The process for the production of MNP is not properly described. Maybe a layout sketch could be added. The only definition as co-precipitation is very generic. Please, be more precise in this definition. And provide some references, that are missing in this case.

Did you try to produce MNP at a temperature different than room temperature? If yes, with which results? Or, if not, why you choose not to modify the temperature? Please add explanations to clarify your point.

At the beginning of page 7, drawbacks should be one word.

Table 1 should be inserted into the introduction, since it does not represent a result, but just a comparison among natural and chemical synthetized MNP.

Figure 2b. regarding the particle size distribution comparison, could you please add a legend on this diagram? In order to identify each PSD with the proper sample in an immediate manner.

Table 2. could you please add technical information about the use of standard deviations for TEM measurements and without the use of SD in the other cases?

DLS size is very different from the other measurement media. How can you deeply explain this data?

Figure 4. why the standard deviation range starts from negative data at 200 ug/mL. Does this have sense in total iron uptake?

Paragraph 3.1.4. could you add mV next to the zeta potential value?

Author Response

We thank the reviewer for their time and support for the research and the paper. We thank the reviewer for their suggestions to improve the manuscript. We have taken these onboard and corrected as outlined below (responses in bold):

An abbreviation list could be a good way to summarize all the acronyms indicated in this paper.

An abbreviation list has now been included at the beginning.

In the abstract, I would suggest to add the standard deviations in the definition of the mean size of nanoparticles, in the abstract section. In this way, a more precise definition of the produced samples can be given to readers.

We are constrained by text limits, but would be more than happy to add this if the editor would allow us top expand the abstract in this way.

The introduction could be enlarged and improved. For example, a brief paragraph containing description and comparison of drug carriers among them could be added (liposomes, magnetosomes, cubosomes, niosomes etc.) In the introduction, a brief general comment on the external stimuli drug delivery systems could be added, with the proper references.

As suggested we have now expanded the introduction especially to include a discussion of current nanoparticle technologies, and external stimuli drug delivery systems.

The total absence of numbered lines made it difficult to provide a proper and fast peer review process.

We apologies. We used the template as provided. We are not sure why the numbered lines were not included. We have added then in now.

The process for the production of MNP is not properly described. Maybe a layout sketch could be added. The only definition as co-precipitation is very generic. Please, be more precise in this definition. And provide some references, that are missing in this case.

We are unsure what improvements the reviewer requires here. The process for the production of MNP will clearly described in the methods. Lines 166-174 give the full preparation. It also clearly describes the ratio we use for our co-precipitation synthesis.

Did you try to produce MNP at a temperature different than room temperature? If yes, with which results? Or, if not, why you choose not to modify the temperature? Please add explanations to clarify your point.

Within our lab, we do routinely produce MNPs via different methods and at different temperatures. However the theme of this special edition is biomimetic MNPs and as such we focused only on MNPs that could be produced under ambient and natural conditions. We thank the reviewer to drawing to our attention that we have not made this point clearly. We have thus added the following sentence to the introduction to clarify this point.

A key aspect to be retained from the natural process is the environmentally friendly processing. “

At the beginning of page 7, drawbacks should be one word.

This has been corrected

Table 1 should be inserted into the introduction, since it does not represent a result, but just a comparison among natural and chemical synthetized MNP.

On your suggestion it has been moved to the introduction.

Figure 2b. regarding the particle size distribution comparison, could you please add a legend on this diagram? In order to identify each PSD with the proper sample in an immediate manner.

This has been added.

Table 2. could you please add technical information about the use of standard deviations for TEM measurements and without the use of SD in the other cases?

DLS does not routinely give SD quantities, but the spectrum are shown in supplementary figure S2. Similarly, the Scherrer equation doesn’t produce SD as it is an estimate calculation.

DLS size is very different from the other measurement media. How can you deeply explain this data?

The reviewer is correct, DLS is a very different sort of measurement giving the hydrodynamic size rather than the particle size. The DLS size measurement is affected by agglomeration of particles and as particle are not monodispersed higher weighting is given to larger particles resulting in a higher size measurement. This information is in the text. “DLS measurement gives the hydrodynamic size of the particle as it appears in solution, and so can be affected by factors such as surface charge and agglomeration. Due to there being a range of particle sizes (not monodispersed) this can also lead to an increase in the size observed by DLS due to the increased weighting given to larger particles in DLS.”

Figure 4. why the standard deviation range starts from negative data at 200 ug/mL. Does this have sense in total iron uptake? This is due to there being a large variation in the results

The 200 ug/ml data point was reassessed and the error were too great for it to be found to be reliable. The particles could not be properly washed from the cells and this resulted in poor data. Thus this data point was removed and the problems with higher concentrations of particles was explained in the text. “The cells were incubated with concentration of particles below 0.2 mg/ml. MNP concentrations of 0.2 mg/ml and above lead to aggregation and difficulty removing residual/surface MNPs giving unreliable data with larger errors. The data shows that particle size in this range…”

Paragraph 3.1.4. could you add mV next to the zeta potential value?

This has been added.

Round 2

Reviewer 1 Report

Thank you for sending the revised version of the manuscript. It contains all the changes mentioned in the author's reply to the first review. It seems that the authors addressed all the issues signaled in my first review. Although t I do not agree with some statements of the authors ( "some larger particles being obscured due to aggregation on the grid, and such particles dominating the XRD signal" in my opinion the smaller particles dominated the XRD signal and lead to larger widths) I believe that the authors have the right to a different opinion as they assume it. There is a small mistake in the revised version, in figure 1b the abscissa axis should be Diameter (nm) as in the original version. I also suggest adding in the legend of this figure that the curves are fitting curves of the size distribution histograms (presented in the supplementary file) and eventually to indicate the fitting function. As such I suggest accepting the paper for publication in the present form (provided the above-mentioned error is corrected).

Author Response

We thank the reviewer for their time reviewing our paper, and appreicing our corrections. We are very grateful for noticing the copy and paste error in figure 2. This has been corrected. Thank you also for the suggested wording for the figure caption. This has also been included. While doing these correction we also noticed a couple of numbering errors in the methods, which we also corrected and added in the joint first authorship statement on the front page. These have all bee shown in read on one version of the manuscript.

The Point the reviewer raises about the XRD is interesting. After re-reading this sentence we thought it could be clear so have slightly changed it. Unfortunately these changes are not in favour of the reviewers opinion. While we agree that visually, the appearance of the XRD spectrum is dominated by smaller particles with wider peaks, the peak height of these is low. It is the larger particles that result in not only narrower peaks, but also higher peaks and it is these that are used to calculate the size using the scherrer equation. Thus it is the larger particles that dominate the calculation. Conversely, 1 large particle will have a larger volume so will diffract much more than 1 small particle. However, in the TEM measurements these are both equal. We have thus rewritten the sentence to capture these points concisely and more clearly. We have also added in a preceding sentence noting that the coatings should not affect the XRD crystallite size as they do not form part of the crystal. Lines 337-341 now read

“Note coating should be invisible to XRD, so sizes for the all the cMNP group, and rMNP group should be similar, regardless of coating or lack of coating. Differences between the observed size of uncoated MNPs via TEM and the size determined via the Scherrer equation may be due to a number of factors such as a lower number (but larger volume) of larger particles making a smaller impact on TEM measurements but dominating the XRD signal.”

Reviewer 2 Report

Now I checked the revised version. It can be published.

Author Response

We thank the reviewer for their time and expertise in reviewing our paper.